# UK alcohol consumption during the COVID-19 pandemic: The role of drinking motives, employment and subjective mental health

Rebecca Louise Monk[1,2]*, Adam W. Qureshi[1,2], George B. Richardson[3], Derek Heim[1,2]

1 Faculty of Arts and Sciences, Department of Psychology, Edge Hill University, Ormskirk, United Kingdom, 2 Liverpool Centre for Alcohol Research, Liverpool, United Kingdom, 3 School of Human Services, College of Education, Criminal Justice, and Human Services, University of Cincinnati, Cincinnati, United States of America

☯ These authors contributed equally to this work.
* monkre@edgehill.ac.uk

**Data Availability Statement:** All relevant data are within the paper and its Supporting Information files.

## Abstract

### Background

Previous investigations suggest that the COVID-19 pandemic effects on alcohol consumption were heterogenous and may vary as a function of structural and psychological factors. Research examining mediating or moderating factors implicated in pandemic-occasioned changes in drinking have also tended to use single-study cross-sectional designs and convenience samples. Aims: First, to explore structural (changed employment or unemployment) and psychological (subjective mental health and drinking motives) correlates of consumption reported during the COVID-19 pandemic using a UK nationally representative (quota sampled) dataset. Second, to determine whether population-level differences in drinking during the COVID-19 pandemic (versus pre-pandemic levels) could be attributable to drinking motives. Method: Data collected from samples of UK adults before and during the pandemic were obtained and analysed: Step1 carried out structural equation modelling (SEM) to explore data gathered during a period of social restrictions after the UK's first COVID-19-related lockdown (27 August-15 September, 2020; n = 3,798). It assessed whether drinking motives (enhancement, social, conformity, coping), employment and the perceived impact of the pandemic on subjective mental health may explain between-person differences in self-reported alcohol consumption. Step 2 multigroup SEM evaluated data gathered pre-pandemic (2018; n = 7,902) in concert with the pandemic data from step 1, to test the theory that population-level differences in alcohol consumption are attributable to variances in drinking motives. Results: Analyses of the 2020 dataset detected both direct and indirect effects of subjective mental health, drinking motives, and employment matters (e.g., having been furloughed) on alcohol use. Findings from a multigroup SEM were consistent with the theory that drinking motives explain not only individual differences in alcohol use at both time points, but also population-level increases in use during the pandemic. Conclusion: This work highlights socioeconomic and employment considerations when seeking to understand COVID-19-related drinking. It also indicates that drinking motives may be

**Funding:** The author(s) received no specific funding for this work.

particularly important in explaining the apparent trend of heightened drinking during the pandemic. Limitations related to causal inference are discussed.

## Introduction

The COVID-19 pandemic not only focussed the minds of those involved in the delivery of public health; it also fuelled much research activity towards elucidating the socioeconomic and (mental) health impacts of this unprecedented global watershed. One area of concern has centred on alcohol-related behaviours. During periods of strict lockdown, on-trade consumption shifted to private, unregulated settings [1] and the description of the pandemic and alcohol as a 'dangerous cocktail' [2] appears apt given that alcohol-related deaths increased by 19% in the UK during 2020 [1]. It is therefore necessary to understand how pandemic alcohol consumption may have varied as a function of structural and psychological factors, and for research to move beyond the preponderance of cross-sectional 'snapshot' studies relying on small convenience samples.

As countries around the globe attempted to control the spread of the virus by enacting a diverse array of measures designed to limit social contact, growing evidence from around the world emerged regarding the relationship between the pandemic and alcohol consumption. In the UK, reductions in on-trade consumption appeared to be offset by increases in reported drinking at home [3], indicating that there were changes in drinking contexts but not necessarily in quantities, which was also observed elsewhere in the world [4, 5, though see 6]. As such, a complex picture of alcohol consumption during the Covid-19 pandemic has emerged.

A body of work also suggests that the impact of the pandemic on consumption has been heterogeneous [see 7 for a review]. Recent meta-analyses indicate that while some people increased their drinking, a largely equivalent percentage reported drinking [8], with the exception of those whose pre-COVID levels of consumption were problematic, where use frequently remained high or increased [9]. Similarly, in the UK, 48% of respondents reported drinking about the same, 26% consumed less, and 26% drank more than usual over the past week [10], although a large-scale cross-sectional online survey in 21 European countries identified decreases in consumption (driven by reduced frequency of heavy episodic drinking events) in all countries, except Ireland and the UK [11]. In short, what emerges is a somewhat mixed literature on national patterns of consumption. Accordingly, there has been a growing focus on the individual and contextual factors that may offer insights into why some people may have consumed more alcohol during the pandemic, while others appeared to maintain pre-COVID-19 drinking patterns.

One etiological explanation for elevated alcohol consumption, already established in pre-COVID times, centres on the experience of elevated stress being a correlate of (excessive) alcohol use [12], and there is evidence that elevated stress and mental health concerns were a feature of life during the pandemic. In Australia, for example, [13] found that close to 80% of respondents in a large (n = 5,070) general population (online) sample reported deteriorations in mental health since the pandemic began, and while this work did not examine whether deteriorated mental health was associated with elevated alcohol consumption, self-reported experiences of stress have been shown to be linked to elevated consumption [14]. Likewise, research in the UK has linked increases in alcohol consumption during the pandemic to poorer/deteriorating mental health [15, 16] and anxiety disorders [10], mirroring findings from multinational research [17–20].

Shedding further light on the potential relationship between the pandemic, experiences of stress and alcohol consumption, recent longitudinal research in the UK found coping motives, but not anxiety, to be significantly associated with increased pandemic drinking over a three-month period [21]. The apparent link between drinking to cope and increased pandemic drinking has also been found in surveys of Belgian college students [22] and US young and middle adults [23] and accords with the notion that drinking to cope with life-stressors (or boredom; [23] may have mediated consumption in response to the pandemic. As such, it may be that the apparent effect of experiences of stress or anxiety on consumption [21] are mediated by variation in alcohol-related beliefs. Indeed, this research links theoretically with the self-medication hypothesis [24] that posits that alcohol is used as a means of improving low mood and/or ameliorating negative affective state or mental/physical states (although see [25], which suggests that a model of affect intensity regulation may be a more advantageous way of understanding the alcohol-mood nexus, rather than theorising about mood valence). It also aligns with pre-COVID research by [26] indicating that coping, enhancement, and conformity, but not social, motives were associated with problem alcohol consumption in the UK. This study also found that coping motives in people with working class backgrounds were related to elevated alcohol consumption. Since COVID-19 exerted disproportionate pressures on those from comparatively disadvantaged socioeconomic backgrounds and in more precarious jobs [27–29], and given that the pandemic shifted routinised contexts and boundaries governing consumption [30], further research examining the role of drinking motives, employment uncertainties and alcohol consumption during the pandemic is warranted.

COVID-19 led to significantly altered employment patterns, home-schooling and associated child-care concerns [31], as well as unemployment [32], all of which have been linked to alcohol consumption. Being younger in age [10, 15, 32], on a higher income [10], having lower educational attainment, problematic pre-pandemic consumption, drinking alone at home [21], and being female (versus male [10, 18], though C.f. [21]) are also variable factors that have been associated with increases in alcohol consumption. In summary, what emerges from the literature is that overall alcohol consumption patterns during the COVID pandemic (and associated lockdowns) were variable and a myriad of interacting factors may mediate or moderate increases/decreases/stasis in drinking behaviours. However, less work has examined how alcohol behaviours may vary as a function of both structural factors associated with work (i.e., changed employment or unemployment) and psychological drivers (such as drinking motives). Much research in this area has also used convenience samples and work using larger and more representative samples is needed.

Towards filling these gaps and providing greater evidence upon which to develop population-level alcohol-related interventions, the present research conducts secondary analyses of data from two large representative samples of UK adults to assess the impact of the pandemic on alcohol consumption. Our approach consists of two steps with associated aims. Step 1 aimed to examine between-persons differences in alcohol consumption to explore factors that may explain divergences in self-reported alcohol consumption. As such, here we examine correlates of pandemic drinking during a period of COVID-19 related social restrictions (between 27 August and 15 September 2020; final n = 3,798) and, using structural equation modelling (SEM), test the theory that COVID-19-related employment and the perceived impact of COVID-19 on one's mental health (subjective mental health) impact self-reported alcohol consumption (AUDIT-C) via drinking motives. Step 2 aimed to explore the mean differences in consumption at the population-level (between two timepoints) and assess whether these may be driven by specific factors. Accordingly, here, we move beyond previous research largely based on single-study cross-sectional designs by leveraging UK nationally representative data gathered before and during the pandemic to (a) test for heterogeneity in pandemic-occasioned

change in alcohol consumption, as well as (b) test the theory that changes in drinking motives may explain pandemic-occasioned differences in consumption. Specifically, we took data gathered in 2018 (n = 7,902) and assessed these against the data from 2020 (represented in step one), during a period in which the UK had lifted full lockdown restrictions but maintained controls on social interactions (n = 3,798).

## Method

### Study population

Two online surveys were carried out by YouGov for the Drinkaware Trust through online panels. Data were obtained using a quota sampling approach designed to collate representative samples. YouGov's research panel consists of over 1,000,000 people in the UK. We declare no conflicts of interest in the use of this data but wish to be very explicit that the data underpinning the analyses were collected by YouGov for the Drinkaware Trust which is an industry-funded body. Neither YouGov or Drinkaware played any role in influencing the research questions we sought to address using this dataset, shaping the analyses or influencing the drafting of the manuscript in any way.

For this research, members were selected based on known demographic characteristics (specifically age, gender, social grade and region), and the sample surveyed was representative of the four nations of the UK. More specifically, predefined quotas were obtained based on the known population profile of adults aged 18–85 years according to gender, age, social grade, and region. The final data were weighted to reflect this profile [33]. The first recorded data from 8,906 UK adults aged 18 to 85 years between 14th May and 5th June 2018. The second contained 9,046 UK adults aged 18 to 85 between 27 August and 15 September 2020. The age breakdown of the sample is shown in Table 1, with the proportion of men in the 2018 sample at 49.1% and 49.6% in the 2020 sample.

### Measures

With respect to the current study, the measures used were:

The Alcohol Use Disorder Identification Test- C (AUDIT-C; [34]) consists of the three consumption questions from the full alcohol use disorders identification test (AUDIT), with each item being scored on a continuous scale from 0–4 (e.g., How often do you have a drink containing alcohol? response options: Never, month or less, 2–4 times per month, 2–3 times per week, 4 + times per week). This is used to supply a measure of an individual's alcohol consumption ($\Omega$ = .78 in both 2018 and 2020), with higher scores indicating greater levels of consumption.

The Drinking Motive Questionnaire Revised Short Form (DMQ-R SF; [35], the short form of the DMQ-R) is a 12 item questionnaire which assesses an individual's motivations for drinking (e.g., because it helps you enjoy a party) on a 5 point likert scale (1, almost never– 5,

**Table 1. Age group for each sample (N (%)).**

|         | 2018          | 2020          |
|---------|---------------|---------------|
| 18–24   | 1039 (11.7%)  | 1090 (12.0%)  |
| 25–34   | 1576 (17.7%)  | 1590 (17.6%)  |
| 35–44   | 1490 (16.7%)  | 1515 (16.7%)  |
| 45–54   | 1636 (18.4%)  | 1652 (18.3%)  |
| 55–64   | 1325 (14.9%)  | 1339 (14.8%)  |
| 65–75   | 1210 (13.6%)  | 1213 (13.4%)  |
| 76+     | 628 (7.1%)    | 646 (7.1%)    |

almost always). Responses can be divided into motivations to drink for enhancement, social, conformity and coping purposes, with higher scores relating to higher endorsements of each respective drinking motive (For Enhancement $\Omega$ = .82 (2018) and .81 (2020); for Social $\Omega$ = .90 (2018) and .89 (2020); for Conformity $\Omega$ = .77 (2018) and .79 (2020); and for Coping motives $\Omega$ = .85 (2018) and .84 (2020)).

For solely the 2020 survey, the following questions relating to COVID-19 life disruption were also asked: Are you concerned about your job security? (response options for selection were as follows: not at all concerned, not very concerned, fairly concerned, very concerned: variable name: "job security"). How much of an impact do you think that the pandemic has had on your mental health and wellbeing? (response options given: to a very large extent, to a large extent, to a moderate extent, to a small extent, to a very small extent: variable name: "subjective mental health"). Are or have you been furloughed? (response options provided: currently furloughed, flexible furlough, was furloughed but now back at work, not been furloughed: variable name "furlough"). To what extent do you agree/disagree with the following statement: "My job has become more stressful due to the Coronavirus pandemic"? (response options supplied: strongly disagree, disagree, neither agree nor disagree, agree, strongly agree; variable name: "job stress"). Where were you mainly working during lockdown (at home, at my usual place of employment, other location(s) outside the home; variable name: "work location")? Did your place of work change at the start of lockdown? (response options: No–I work in the same office/site/place of work as I did previously, Yes–I started to work from home but didn't previously, Yes–my employer moved me to a different office/site/place of work as I did previously: variable name "work location change").

We also included social grade, gender, and age as covariates in the analyses. Social grade was recorded binomially (ABC1 (upper middle class, middle class, lower middle class) x C2DE (skilled working class, working class, lower level of subsistence reference category–C2DE); reference category C2DE). Social grade responses were coded based on self-report. Respondents were asked who the chief income earner in their household was, and then asked follow-up questions about their job (e.g., self-employed or employed, sectors, seniority). The responses were then coded (by YouGov) to produce the ABC1 and C2DE (A = Higher managerial, administrative and professional, B = Intermediate managerial, administrative and professional, C1 = Supervisory, clerical and junior managerial, administrative and professional, C2 = Skilled manual workers, D = Semi-skilled and unskilled manual workers, E = State pensioners, casual and lowest grade workers, unemployed classifications broadly). Gender was classified as male and female (male = reference category), and age was analysed continuously.

**Principal Components Analysis (PCA).** Before conducting our SEM analyses, we used PCA to construct summaries of the life disruption variables due to moderate to large (rs = .1 to 6) correlations and overlapping content among them. Variables of furlough, work location, work location change, job security, job stress and subjective mental health were entered into a PCA based on the correlation matrix, using IBM SPSS Statistics 26. Once we identified an acceptable solution, we used Bartlett's methods to save component scores for use in our SEM.

The Kaiser-Meyer-Olkin (KMO) measure of sampling adequacy was mediocre (.515), though Bartlett's Test of Sphericity was significant (X2 (21, N = 3630) = 3121.693, p < .001). The initial scree plot shows a drop in the percentage of explained variation after component three. If each variable contributed equally, they would contribute 16.67% to the total variance (indicated by the line in Fig 1). This also suggested that the focus of analyses should be on the first three components.

This results in 27.92% of the variance being explained by component 1, 25.21% by component 2 and 19.65% by component 3, for a total of 72.78%. The loadings are shown in the pattern matrix in Table 2 below.

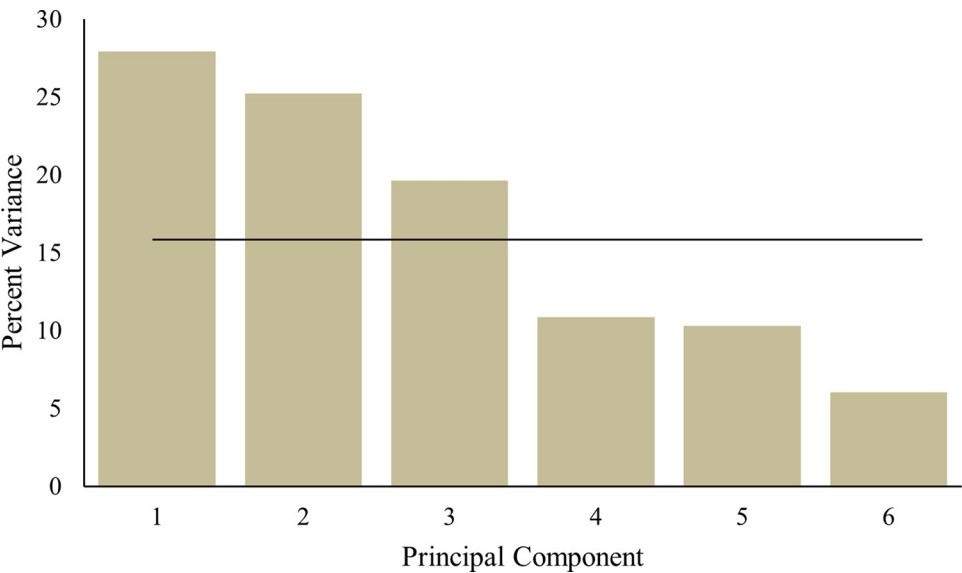

**Fig 1. Scree plot of principal component analysis (PCA).**

The loadings suggest that the variables that contribute the most to PC1 were respondents' work location during the pandemic and if that location changed. Therefore, PC1 was named "Employment location", with higher scores indicating working in the same place as before the pandemic.

Variables contributing to PC2 were the perceived impact of COVID-19 on mental health (subjective mental health) and if participants reported that their jobs had become more stressful due to the pandemic. The pattern here suggests that the higher the reported stress, the more detrimental the perceived impact of Covid-19 on respondents' mental health. Consequently, we labelled this component was labelled "subjective mental health". Higher scores were associated with higher covid-19 related stress and greater perceived damage on mental health.

Finally, PC3 consisted of questions relating to furlough (whether they had been furloughed) and if respondents were concerned over their job security, with those more concerned about job security more likely to have been or currently furloughed. This was therefore labelled "Employment security", with higher scores indicating more job security and not being furloughed.

These three summary measures of the life disruption variables were then entered into the SEM model for our initial aim (examination of between-persons differences in alcohol consumption, with a focus on COVID-19 related life disruptions and drinking motives).

**Table 2. Pattern matrix showing loadings of variables on the extracted principal components.**

| *PATTERN* | PC1 | PC2 | PC3 |
|---|---|---|---|
| Work location | **.895** | .058 | .019 |
| Work location change | **.900** | -.014 | .080 |
| Job stress | .161 | **.837** | .150 |
| Mental health | .136 | **-.739** | .176 |
| Furlough | -.111 | .200 | **.863** |
| Job security | .072 | .284 | **-.719** |

## Analyses

This study used SEM to examine the associations among COVID-19-related employment and subjective mental health and wellbeing (as assessed through subjective self-report), drinking motives, and alcohol use. We used Amos software, the maximum likelihood (ML) estimator, conducted significance testing at the $\alpha$ = .05 level, and interpreted $p$-values associated with unstandardized estimates. Results of Little's MCAR tests ($p$s > .05) did not lead us to reject the hypothesis that the missing data were missing completely at random (MCAR), suggesting list-wise deletion was appropriate. Therefore, we excluded 5,248 (58%) participants from the 2020 dataset and 1,004 (11%) from the 2018 dataset due to missingness. Specifically, 1,004 respondents in the 2018 dataset reported not drinking alcohol and hence had no drinking motive data either. For the 2020 dataset, 1,221 (14%) also reported not drinking alcohol and had no drinking motives data. Additionally, a further 4,027 (45%) had no data for the furlough and job security questions, and of those 3,412 also had no job stress or workplace information data. The final sample for preliminary analysis was n = 3,798 for the 2020 dataset and n = 7,902 for the 2018 dataset, with all participants having completed values for AUDIT-C, gender, age, social grade, drinking motives and in the case of the 2020 dataset, questions concerning the COVID-19 life disruption questions.

We employed conventional and multigroup SEM to achieve our first and second aims, respectively. Importantly, multigroup SEM allowed us to determine, using invariance testing, whether variance in alcohol use increased at the population level during the pandemic, consistent with previous evidence of heterogenous effects on use; as well as whether differences between 2018 and 2020 might be attributable to population-level differences in drinking motives. We carried out three stages of structural invariance testing: 1.) invariance of structural regression coefficients, 2.) scalar invariance (i.e., equivalence of the intercept) and 3.) homogeneity of residual variances. Invariance of the structural effects provides evidence of equal associations among independent and dependent variables across groups. This type of invariance suggests there are no group-by-predictor interaction effects and therefore, one-unit changes on independent variables have the same meaning in terms of dependent variable changes in both groups. Invariance of intercepts mean the origins of the slopes are equal, which implies that mean group differences in dependent variables may be attributable to differences in independent variables between groups. In contrast, detected intercept non-invariance implies that independent variables do not fully explain main effects of group on a dependent variable. If invariance of residuals also holds, this implies that both modelled and unmodeled sources of variance in dependent variables are likely to be equivalent across the groups.

Multigroup SEM has many advantages over multiple regression-based testing of group-by-independent variable interaction effects. Whereas the regression-based approach only estimates an intercept for the reference group, multigroup SEM allows researchers to estimate intercepts for all groups and test them for equality. Multigroup SEM also allows researchers to estimate residuals for all groups and test them for equivalence; when these are equal, all sources of variance in dependent variables may be homogenous across two groups and standardized estimates are identical between them.

In the current study, heterogeneity in pandemic-occasioned differences in drinking due to modelled independent variables (e.g., pandemic-occasioned increases in drinking could be larger in those who endorsed greater coping motives pre-pandemic) is detected as non-invariance in effects on alcohol use, while heterogeneity in pandemic-occasioned change in drinking due to unmodelled variables is detected as intercept and/or residual non-invariance. That is, differences on the population mean due to unmodelled factors would appear as an unexplained increase in the pandemic sample intercept relative to the pre-pandemic sample,

whereas differences limited to certain age or other groups would likely increase the variance in the former group relative to the latter.

## Hypothesized model

*Step 1: Between-persons differences in alcohol consumption to assess explore factors that may explain differences in self-reported alcohol consumption*

To achieve our first aim, we used conventional SEM and data from 20FMCAR20 (n = 3,798). Drawing on the literature, we theorized that drinking motives (enhancement, social, conformity, coping) translated employment as well as subjective mental health and well-being into alcohol use (AUDIT-C). We also theorized that age, gender, and social grade (ABC1 x C2DE) were exogenous. We encoded these theoretical assumptions into our SEM and tested it.

*Step 2: Mean differences in consumption at the population-level (between two timepoints) and if these are driven by specific factors.*

To achieve our second aim, we used multigroup SEM and data from both 2018 (n = 7,902) and 2020 (n = 3,798). We theorized that drinking motives impacted alcohol use and that the same demographic variables were exogenous as in the model previously described. We encoded these theoretical assumptions into our multigroup SEM and imposed the invariance testing constraints previously described. As a reminder, the COVID-19-related employment and subjective mental health variables did not appear in the 2018 survey because the pandemic had not yet occurred; therefore, these variables were not included in the multigroup SEM.

## Model fit

The current study used a variety of indices to obtain a robust assessment of model fit. We considered the substantive meaningfulness of the model and regarded Tucker-Lewis (TLI) and comparative fit (CFI) indices greater than or equal to .95 [36, 37], along with root mean square error of approximation values less than or equal to .05 (RMSEA [38]) as evidence of acceptable fit to the data. We also considered significant $\chi2$ likelihood ratio statistics as evidence that the hypothesis of exact fit should be rejected [22]. When testing for invariance, changes in CFI of less than -.01 between models were regarded as evidence of non-invariance [39].

## Results

### Step 1: 2020 data

Prior to our analyses, we checked the dataset for univariate outliers and examined pairwise plots for any heteroscedasticity. Multivariate outliers were checked as per [40], resulting in 24 participants being excluded from an original sample of 3,798. The final sample was therefore 3,774. This dataset was also used for the PCA analysis.

### SEM

The initial model had paths from age, gender and social grade to all three components of life disruption [Employment location, Subjective mental health, Employment security], and paths from these predicting each of the drinking motives, which in turn predicted alcohol consumption (see S1 Table in S1 File). Covariances were also present between all drinking motives (as per [26]).

In the second model those paths that were not significant were removed, which did not significantly affect model fit (CFI = .000, p = .573); S2 Table in S1 File). In the final model covariances and direct paths that significantly improved the model fit (as per modification indices; see Table 3 and Fig 2) were added (CFI = .076, p < .001). No covariances between life

**Table 3.** Final model parameters.

| Regression Weights | | | Estimate | S.E. | *p* | β |
|---|---|---|---|---|---|---|
| employment location | <— | Social Grade | -.651 | .031 | *** | -.317 |
| Mental health | <— | Social Grade | .056 | .033 | .085 | .027 |
| employment security | <— | Social Grade | .126 | .033 | *** | .061 |
| employment location | <— | Gender | -.144 | .031 | *** | -.072 |
| Mental health | <— | Gender | .301 | .032 | *** | .150 |
| employment security | <— | Gender | | | | |
| employment security | <— | Age | .003 | .001 | .012 | .041 |
| Mental health | <— | Age | -.011 | .001 | *** | -.139 |
| employment location | <— | Age | .007 | .001 | *** | .095 |
| Enhancement | <— | employment location | -.040 | .016 | .010 | -.038 |
| Social | <— | employment location | -.030 | .015 | .053 | -.028 |
| Conformity | <— | employment location | | | | |
| Coping | <— | employment location | | | | |
| Coping | <— | employment security | -.096 | .012 | *** | -.112 |
| Conformity | <— | employment security | -.055 | .01 | *** | -.080 |
| Social | <— | employment security | | | | |
| Enhancement | <— | employment security | | | | |
| Coping | <— | Mental health | .181 | .014 | *** | .213 |
| Conformity | <— | Mental health | .044 | .011 | *** | .065 |
| Social | <— | Mental health | .032 | .018 | .072 | .029 |
| Enhancement | <— | Mental health | .033 | .017 | .056 | .031 |
| Coping | <— | Age | -.004 | .001 | *** | -.063 |
| Conformity | <— | Age | -.009 | .001 | *** | -.173 |
| Social | <— | Age | -.018 | .001 | *** | -.214 |
| Enhancement | <— | Age | -.016 | .001 | *** | -.191 |
| Conformity | <— | Gender | -.137 | .021 | *** | -.100 |
| Enhancement | <— | Gender | -.198 | .032 | *** | -.093 |
| Social | <— | Gender | -.194 | .033 | *** | -.089 |
| Coping | <— | Social Grade | -.068 | .024 | .006 | -.039 |
| AUDIT-C | <— | Enhancement | .957 | .049 | *** | .368 |
| AUDIT-C | <— | Social | .226 | .049 | *** | .088 |
| AUDIT-C | <— | Conformity | -.385 | .063 | *** | -.095 |
| AUDIT-C | <— | Coping | .755 | .051 | *** | .232 |
| AUDIT-C | <— | employment security | .082 | .038 | .031 | .030 |
| AUDIT-C | <— | Age | .019 | .003 | *** | .089 |
| AUDIT-C | <— | Gender | -.743 | .076 | *** | -.134 |
| **Covariances** | | | | | | |
| Gender | <—> | Age | -.536 | .104 | *** | -.084 |
| err15 (Mental health) | <—> | err16 (employment security) | -.098 | .016 | *** | -.101 |
| err14 (employment location) | <—> | err16 (employment security) | .055 | .015 | *** | .059 |
| err13 (Enhancement) | <—> | err11 (Coping) | .346 | .015 | *** | .406 |
| err10 (Conformity) | <—> | err11 (Coping) | .168 | .009 | *** | .308 |
| err9 (Social) | <—> | err10 (Conformity) | .288 | .012 | *** | .410 |
| err13 (Enhancement) | <—> | err9 (Social) | .691 | .021 | *** | .628 |
| err13 (Enhancement) | <—> | err10 (Conformity) | .164 | .012 | *** | .236 |
| err9 (Social) | <—> | err11 (Coping) | .275 | .015 | *** | .319 |
| **Variances** | | | | | | |

*(Continued)*

**Table 3.** (Continued)

| Regression Weights | | | Estimate | S.E. | p | β |
|---|---|---|---|---|---|---|
| Social Grade | | | .238 | .005 | *** | |
| Gender | | | .250 | .006 | *** | |
| Age | | | 162.657 | 3.745 | *** | |
| err14 (employment location) | | | .885 | .020 | *** | |
| err15 (Mental health) | | | .954 | .022 | *** | |
| err16(employment security) | | | .994 | .023 | *** | |
| err13 (Enhancement) | | | 1.084 | .025 | *** | |
| err9 (Social) | | | 1.115 | .026 | *** | |
| err10 (Conformity) | | | .444 | .010 | *** | |
| err11(Coping) | | | .670 | .015 | *** | |
| err12 (AUDIT-C) | | | 5.344 | .123 | *** | |
| **Squared Multiple Correlations** | | | | | | |
| Mental health | | | .046 | | | |
| employment security | | | .005 | | | |
| employment location | | | .116 | | | |
| Coping | | | .073 | | | |
| Conformity | | | .051 | | | |
| Social | | | .054 | | | |
| Enhancement | | | .047 | | | |
| AUDIT-C | | | .304 | | | |

***p < .001

disruption variables were suggested. This final model showed excellent fit (e.g., $\chi^2 = 1.63(15)$, p = .059; see, S2 Table in S1 File).

## Direct effects

Changes in employment location were associated with higher endorsement of both social and enhancement motives, while greater employment security was positively associated with

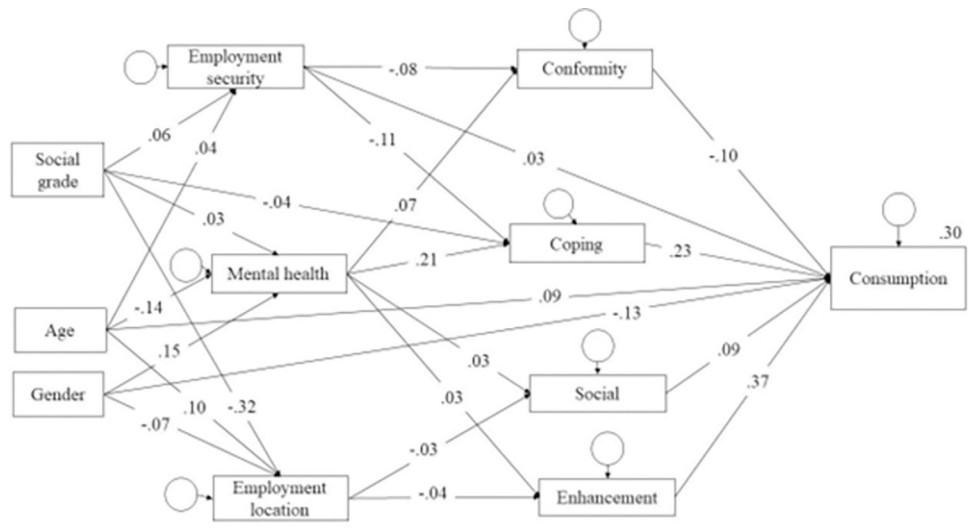

**Fig 2. Final model diagram (values are standardised estimates).**

coping and conformity as well as lower alcohol consumption. Those reporting greater impact on their mental health and wellbeing showed higher endorsement of all drinking motives.

Higher endorsement of enhancement, coping and social motives were also all associated with higher consumption. However, higher endorsement of conformity motives was linked to lower consumption.

Turning to our covariates, the model suggests the following: those having more employment insecurity tended to be of a lower social grade and also younger. Women, younger adults, and those of a higher social grade were all more likely to have experienced changes in their employment location (e.g., moved from workplace to working from home). Those experiencing greater subjective impacts on their mental health and wellbeing tended to be women and younger adults.

Men showed higher social, enhancement and conformity motives (though there was no difference in coping motives between men and women), while increasing age was associated with lower endorsement of all motives. Lower social grade was linked to higher coping motives. With regards to alcohol consumption, older adults and men tended to have higher consumption (see Fig 2 for summary).

### Indirect effects of life disruption components

We tested for indirect effects of the life disruption components, controlling age, gender, and social grade. All indirect effects are shown in Fig 2 (and S3 Table in S1 File), and those related to consumption and the component factors are summarised in more detail. Changes in employment location were indirectly associated with higher consumption via enhancement and social motives. Less employment security was indirectly associated with higher consumption via conformity motives, though having more employment security was indirectly associated with higher consumption via coping motives. Higher self-reported (subjective) impact on mental health and well-being was indirectly associated with higher consumption via coping motives, but with lower consumption via conformity motives.

### Step 2: 2020 vs 2018 data

Using multigroup SEM, we tested whether effects of covariates (gender, age group, social grade (ABC1 x C2DE) and drinking motives (enhancement, social, conformity, coping) on alcohol consumption (AUDIT-C) varied between 2018 and 2020.

Comparing the two samples showed that there was no difference in gender mix or social grade between the two time points, but there were differences in age with a higher proportion of older adults' responses included in the 2018 sample. All drinking motives were more highly endorsed in the 2020 sample, and alcohol consumption was also higher (see Table 4 for descriptive statistics, group comparisons and standardised mean difference effect sizes (d)). All model parameters are shown in Supplementary Materials (S4–S8 Tables in S1 File).

After an initial baseline model where parameters were allowed to vary (M1, S4 Table in S1 File), we tested for structural invariance and results suggest invariance held at all levels (all $\Delta$CFI < -.01), indicating that the effects of predictors, their origins, and their relative importance in explaining alcohol consumption were equivalent in the population at the two time points (M2, S5 Table in S1 File). A summary is supplied in Table 5.

Intercept invariance suggested modelled predictors explained the population-level difference in alcohol consumption (M3, Supplementary Materials S6 Table in S1 File). No residual non-invariance was detected, suggesting unmodelled sources of variance in alcohol use were equivalent across the time points. (M4, Supplementary Materials S7 Table in S1 File). Model fit and comparisons are shown Supplementary Materials (S8 Table in S1 File) Alcohol

**Table 4. Descriptive statistics and group comparisons.**

| | | 2018 (*n* = 7902) | 2020 (*n* = 3798) | | Standardised mean difference (SMD; *d* (CI)) |
|---|---|---|---|---|---|
| Gender | Male | 3950 (49.99%) | 1862 (49.03%) | $X^2$ (1, N = 11700) = .949, p = .333 | .02 (-.02, .05) |
| | Female | 3952 (5.01%) | 1936 (5.97%) | | |
| Age Group | 18–24 | 576 (7.29%) | 350 (9.22%) | $X^2$ (6, N = 11700) = 898.15, p < .001 | .58 (.54, .61) |
| | 25–34 | 997 (12.62%) | 917 (24.14%) | | |
| | 35–44 | 975 (12.34%) | 900 (23.7%) | | |
| | 45–54 | 2199 (27.83%) | 933 (24.57%) | | |
| | 55–64 | 1853 (23.45%) | 575 (15.14%) | | |
| | 65–75 | 866 (1.96%) | 105 (2.76%) | | |
| | 76+ | 436 (5.52%) | 18 (.47%) | | |
| Social Grade | ABC1 | 4703 (59.52%) | 2317 (61.01%) | $X^2$ (1, N = 11700) = 2.37, p = .126 | .02 (-.01, .06) |
| | C2DE | 3199 (41.48%) | 1481 (38.99%) | | |
| Enhancement | | 2.32 (1.07) | 2.45 (1.07) | t (11698) = -6.24, p < .001 | .12 (.08, .16) |
| Social | | 2.37 (1.10) | 2.47 (1.09) | t (11698) = -4.48, p < .001 | .09 (.05, .13) |
| Conformity | | 1.41 (.66) | 1.44 (.69) | t (11698) = -2.31, p = .021 | .05 (.01, .08) |
| Coping | | 1.64 (.87) | 1.69 (.85) | t (11698) = -2.49, p = .013 | .04 (.01, .09) |
| AUDIT_C | | 4.68 (2.81) | 4.94 (2.77) | t (11698) = -4.59, p < .001 | .09 (.05, .13) |

consumption appeared to be greater in 2020 compared to 2018, and this difference appears partly attributable to greater endorsement of Enhancement and Coping drinking motives (SMD = .12 and .04 respectively) at the population level. Other factors (age group and social grade) were also significant, but their standardised effect sizes were quite small (< .05). However, consumption was higher in those identifying as men (-.173; men coded as 0, women as 1).

## Discussion

Overall, the current findings add to a by now relatively substantial body of work highlighting the potentially pernicious influence of the pandemic on alcohol consumption behaviours. Using invariance testing and two large UK national datasets collected in 2018 (pre pandemic) and in 2020 (in a period of pandemic-related social restrictions following lockdown), we assessed whether the uncertain context of the pandemic may have been associated with increased alcohol consumption at the population level, and the extent to which drinking motives may have underpinned any such apparent trends. In so doing, we aimed to contribute to an evidence base which to date has been characterised by studies utilising varying sample sizes and measures of alcohol consumption. These previous studies had yielded somewhat inconsistent findings with regards to whether the pandemic increased [e.g., 32], decreased [e.g., 9, 11] or did not impact [e.g., 41] alcohol consumption levels.

The current study suggests that the context of enforced social restrictions may have been associated with elevated consumption levels. While these data were obtained from two population level surveys, the same nationally representative (quota-based) design was used to collect

**Table 5. Model fit statistics.**

| | DF | $\chi^2$ | *p* | NFI | CFI | RMSEA (CI) |
|---|---|---|---|---|---|---|
| Initial model | 24 | 21.78 | < .001 | .920 | .923 | .074 (.069, .080) |
| Model 2 | 28 | 18.77 | < .001 | .919 | .923 | .069 (.064, .074) |
| Final model | 15 | 1.63 | .059 | .996 | .999 | .013 (.000, .022) |

the data in 2018 and 2020, implying that we may infer from our findings that there was a general trend of increased consumption at the population level during the Covid-19 pandemic, which aligns with previous work [32]. Moreover, our invariance testing provides evidence signifying that drinking motives may not only explain some of the differences in alcohol use at the between-persons level, but also at the level of pre- and during-pandemic population means. This finding suggests the observed differences in alcohol consumption between the 2018 and 2020 surveys may be partly attributable to temporal differences in drinking motives. Specifically, we found that increased alcohol consumption in the 2020 sample reflected increases in Enhancement and Coping drinking motives among this group, with a stronger effect of endorsing Enhancement motives relative to Coping motives. Here, our findings are partly consistent with the recent findings regarding the role of coping motives in pandemic consumption [21–23] but also suggest that motives to enhance positive states may have been particularly important in the context of the national social restrictions. This differs somewhat from the findings by [22], who found that enhancement motives were inversely related to consumption. Our analyses also suggest that higher consumption was associated conformity motives (in line with previous research [42], as well as with being male and with lower endorsement motives in the pandemic cohort.

Our invariance analyses also pointed to an absence of heterogeneity in pandemic-occasioned differences in drinking. Specifically, we detected no evidence that the pandemic moderated (buffered or amplified) effects of our modelled predictors on alcohol use, or alternatively, that our modelled predictors moderated pandemic effects on use. We also detected no evidence that unmodelled variables increased the alcohol consumption mean (i.e., equivalence of intercepts held) or of heterogeneous pandemic-related effects on drinking (i.e., equivalence of residuals held). The findings of intercept and residual invariance across the pre- and during-pandemic samples are striking and seem to suggest new causes of drinking did not emerge during the latter period, given that would have likely resulted in an unexplained increase on the alcohol use mean and/or variance. Overall, then, pandemic-occasioned change in drinking seemed to occur only on the population mean and was explained by pre-pandemic causes of drinking such as motives.

Our analyses of the 2020 dataset also suggest that being older was associated with elevated alcohol consumption during the pandemic (during a period in which the UK had emerged from its first period of lockdown but was still under social restrictions). This contrasts with earlier cross-sectional work, reliant on retrospective self-assessments of pre-pandemic drinking, identifying younger people as being at particular risk of elevated consumption [10, 15, 32]. In relation to gender differences in alcohol consumption during the pandemic, current findings suggest that men were more likely to endorse enhancement, coping and social motives which were, in turn, associated with elevated alcohol consumption. Given that previous research has painted a mixed picture with regards to gender differences in pandemic consumption [see 8, 10, 18, 21], our findings shed light on this field and implicate alcohol-related motives as an important explanatory component when seeking to understand gender-based patterns of pandemic drinking.

The current research contributes knowledge regarding how inequality may have been associated with elevated alcohol consumption during the pandemic. In this way, we found changes in employment location were indirectly associated with higher consumption via enhancement and social motives. By documenting that higher levels of alcohol consumption were, in part, associated with changes in employment location, the current research extends previous work highlighting how the pandemic disproportionately impacted people from poorer backgrounds [27, 29]. This is particularly important given that drinking alcohol is a health behaviour associated with manifold adverse outcomes that inordinately impact those who are

socioeconomically disadvantaged [43]. In addition, the present analysis indicates that coping motives may be important in understanding the relationship between socioeconomic status and alcohol consumption, a finding which may have implications for the development of interventions for those most vulnerable within society. Further evidencing the importance of coping motives, both subjective mental health and employment security concerns were indirectly associated with higher consumption via coping motives. Here, those who reported that COVID-19 had had a more negative impact on their mental health and those with a lower sense of security (i.e., feeling that their job was at risk or having been furloughed) reported elevated coping motives which, in turn, were linked to increased consumption. In contrast, fewer employment security concerns were indirectly associated with lower consumption via the lowering of conformity motives. In short, the current analyses contribute to the existing body of work highlighting the potentially significant role of coping motives when seeking to understand patterns of drinking more generally [26], as well as during the COVID-19 pandemic [21].

The current work thereby begins to move beyond the preponderance of 'snapshot' self-report assessments of consumption that have characterised much of the literature on COVID-19-related alcohol consumption. In this way, earlier work tended to encourage respondents to reflect on any changes to their drinking pre versus post lockdown/pandemic and, as such, may have been particularly sensitive to long-documented limitations concerning retrospective sense-making and recall [44]. In contrast, the current work assessed differences in population-level drinking patterns between time points. Of course, future research will need to take a longitudinal approach to better assess whether drinking motives, for instance, caused within-persons, between-persons, and ultimately population-level changes in alcohol consumption drinking patterns between time points.

While the two datasets used in the analyses (2018 and 2020) were relatively large, and invariance testing allows for a number of statistical inferences about differences over time, several study limitations need to be borne in mind when considering findings. First, although our approach (i.e., SEM applied to data from two large cross-sectional samples) provided tests of the theories we encoded into our models, it did not provide testing that is as stringent as in experimental designs. Our models assumed, for instance, that there were no omitted confounders, yet we did not have design features like experimental control that could provide strong assurance this assumption held. Although the lack of differences in the effects of drinking motives between the levels of inter-individual and time point differences is consistent with causation, it remains possible that our estimates are biased by omitted variables that also have effects similar in magnitude at both these levels. Thus, causal inference from our results must remain relatively tentative. Because experimental tests of the theories evaluated here may be infeasible, future research adding a longitudinal perspective seems to provide a logical next step in terms of conducting a more rigorous test of our assumptions. Research tracking longitudinal variability in individual drinking practices would add further insights into how the changing nature of the pandemic over time (i.e. in terms of divergences in the degree/nature of social restrictions etc over time) may also have impacted drinking practices and thus provide insights into within-person changes in drinking.

Second, concerns regarding the veracity of self-report measures [45, 46] should be noted in the context of our reliance on participant response data with possible biases due to shared method variance. Specifically, it should be acknowledged that it is possible that there may be some social-desirable responding in participants' accounts of their social grade and future research may benefit from the use of other indices to obtain a more objective assessment (e.g., Index of Multiple Deprivation recorded based on each individual's local Super Output Area [26]. Furthermore, the measure of mental health and well-being was derived from a single-

item assessment ("How much of an impact do you think that the pandemic has had on your mental health and wellbeing?"), meaning that this is a self-reported subjective account of mental health. While this approach is akin to similar assessments of pandemic drinking and mental health [e.g., 13], subjective mental health and well-being is related but not necessarily analogous with psychological well-being [47]. We therefore caution that the current findings may not reflect research which has used more objective assessments of mental health in the COVID-19 pandemic [e.g., 15] and future research in this regard is recommended.

It is possible that the online nature of the survey could have biased our data towards computer-literate respondents, potentially impacting the representativeness of our findings. It should also be noted that the data used for this study, while large, cannot be held to be fully nationally representative, since they were drawn from an industry-sponsored web panel and this quota sampling is not fully representative in the sense that not every potential participant has a known probability of participation (although this is also routinely the case for the studies which, for example, seek to estimate prevalence of disorders based on national sampling). Further, it should be born in mind that there more data were excluded from the 2020 pandemic dataset (58%) than from the from the 2018 dataset (11%) owing to missingness. While this greater degree of missing data may be explained by additional (and potentially distracting) pressures that people during the pandemic may have been facing when filling out their surveys, these data were missing at random based on Little's MCAR test, and, as such, we did not find any evidence that listwise deletion would introduce bias. Finally, future data collection comparing responses further into the pandemic and again after its peak would be fruitful to better contextualise the current findings.

In conclusion, the current study compared a UK representative (quota sampled) dataset obtained during a period of the pandemic in 2020 (in which social restrictions were in place) with another obtained in 2018. Findings suggest that UK alcohol consumption was greater during this period of the COVID-19 pandemic and that this may be partly attributable to increases in Enhancement and Coping drinking motives. Further, when considering 2020 data only, the study identifies that both employment location and job security were both directly and indirectly (via drinking motives such as coping), associated with elevated alcohol consumption. As such, our research identifies working from home and more precarious employment situations as possible drivers of alcohol consumption during the COVID-19 pandemic. Overall, this study extends previous research documenting that the pandemic magnified socioeconomic disparities in alcohol consumption and suggests that alcohol-related drinking motives may be particularly important in explaining UK drinking during the pandemic.

## Supporting information

**S1 File.**
(DOCX)

**S2 File.**
(XLSX)

## Author Contributions

**Conceptualization:** Rebecca Louise Monk, Adam W. Qureshi, George B. Richardson, Derek Heim.

**Formal analysis:** Rebecca Louise Monk, Adam W. Qureshi, George B. Richardson, Derek Heim.

**Investigation:** Rebecca Louise Monk, Adam W. Qureshi, George B. Richardson, Derek Heim.

**Methodology:** Rebecca Louise Monk, Adam W. Qureshi, George B. Richardson, Derek Heim.

**Project administration:** Rebecca Louise Monk, Adam W. Qureshi, George B. Richardson, Derek Heim.

**Resources:** Rebecca Louise Monk, Adam W. Qureshi, George B. Richardson, Derek Heim.

**Software:** Rebecca Louise Monk, Adam W. Qureshi, George B. Richardson, Derek Heim.

**Supervision:** Rebecca Louise Monk, Adam W. Qureshi, George B. Richardson, Derek Heim.

**Validation:** Rebecca Louise Monk, Adam W. Qureshi, George B. Richardson, Derek Heim.

**Visualization:** Rebecca Louise Monk, Adam W. Qureshi, George B. Richardson, Derek Heim.

**Writing – original draft:** Rebecca Louise Monk, Adam W. Qureshi, George B. Richardson, Derek Heim.

**Writing – review & editing:** Rebecca Louise Monk, Adam W. Qureshi, George B. Richardson, Derek Heim.

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
