## [Decision Letter · Decision Letter 0]

5 Aug 2022

PONE-D-22-17576UK alcohol consumption during the COVID-19 pandemic:  The role of drinking motives, employment and mental health concernsPLOS ONE

Dear Dr. Monk, Thank you for submitting your manuscript to PLOS ONE. After careful consideration, we feel that it has merit but does not fully meet PLOS ONE’s publication criteria as it currently stands. Therefore, we invite you to submit a revised version of the manuscript that addresses the points raised during the review process.

 The data come from an industry sponsored source and although the authors suggest that they are 'nationally representative', no supporting data are provided. Comparisons are made between cross-sectional data collected pre-pandemic (2018) and then during the pandemic in 2020, to assess 'changes'. This is a major weakness of the study, as longitudinal follow up would have been the appropriate method. It is also noted that 5,248 (58%)

participants were excluded from the 2020 dataset and 1,004 (11%) from the 2018 dataset due to missingness. The implications of this loss of information which disproportionately affects persons evaluated during the pandemic has not been addressed.  The paper is verbose and meanders, and as such, its style can be significantly improved. The Introduction can be markedly reduced from the current 6 pages. There are multiple statistical analyses (model fit statistics; indirect effect values; invariance tests) which do not contribute to the overall hypothesis and would be more appropriately included with the Supplementary materials. Removing these tables would improve the focus of the paper. Similarly the Discussion is too long and also lacks clarity and focus.  The reviewers, provide detailed commentary, and the concerns raised by Reviewer 2 must be addressed. 

We look forward to receiving your revised manuscript.

Kind regards,

Anselm J. M. Hennis

Academic Editor

PLOS ONE

Journal Requirements:

3. Please ensure that you include a title page within your main document. You should list all authors and all affiliations as per our author instructions and clearly indicate the corresponding author.

Reviewers' comments:

Reviewer's Responses to Questions

**Comments to the Author**

1. Is the manuscript technically sound, and do the data support the conclusions?

Reviewer #1: Yes

Reviewer #2: Partly

2. Has the statistical analysis been performed appropriately and rigorously? 

Reviewer #1: Yes

Reviewer #2: I Don't Know

3. Have the authors made all data underlying the findings in their manuscript fully available?

Reviewer #1: No

Reviewer #2: Yes

4. Is the manuscript presented in an intelligible fashion and written in standard English?

Reviewer #1: Yes

Reviewer #2: No

5. Review Comments to the Author

Reviewer #1: The article is well written and clear. It would be good to discuss other factors that might be related to increased consumption, such as increased alcohol availability at home, which in turn are known factors to lead to more consumption. Time spent online during the pandemic could be another factor related to increased consumption, but it was never asked. In addition, the authors can better explain how an online survey becomes representative of the general population, when there are differences in access to internet, mobile phones, etc. Finally, the data from 2020 reflect the beginning of the pandemic and not its peak, so this limitation can also be discussed, as we need to have more information afterwards and after the pandemic, to better contextualize the findings and their robustness.

Reviewer #2: This paper aimed to examine the impact of drinking motives and “mental health concerns” on alcohol use in the context of the Covid-19 pandemic in the UK. It used two, independent datasets, one collected well-before the pandemic in 2018 and one collected during the first year of the pandemic. While the datasets are relatively large and the authors attempted to employ a rigorous statistical approach, I lack enthusiasm for the paper as written, and make suggestions that I hope can help improve the manuscript.

Overall

Throughout the manuscript the authors state that they are looking at “change” in alcohol consumption as a result of the pandemic. This not possible given that the data they used is not cohort data. The data allows them to examine *differences* between pre and early pandemic periods, but they cannot assess changes in either alcohol use or in drinking motives without following the same people over time. I suggest reframing the manuscript to reflect this.

Introduction

The introduction does well to provide background on the drinking motives and alcohol use during the pandemic, but it was difficult to follow because it seemed wander between a global perspective and then a UK one. It was hard to tell when the intro was discussing findings that were specific to the UK and when they were from other countries. If putting the UK in a global framework is important to the authors, then I suggest starting with a global perspective then honing in on the specific UK situation.

Also, when the authors note different findings across different countries and the need for additional research to address these differences, the implication is that findings should be consistent across countries. This seems unnecessary since we would expect different reasons for drinking across different cultural contexts. Rather, it would seem that to gain a full picture of the impact of the pandemic on drinking globally, we want rigorous research from all these areas, and not that the findings all have to agree with each other.

No clear reason for why we need this research is presented, that is, how the findings will be used or what they will inform. This is admittedly a pet peeve of mine, but the argument that this work is necessary to “fill the gaps” without explaining why we need to fill these gaps in the first place is a less-than-compelling argument for undertaking this work.

No theoretical framework for examining drinking motives is presented, such as the self-medication hypothesis. Also, the frequent comorbidity between alcohol problems and mental health is not well described in the introduction.

Finally, the aims are unclear and confusing as written. They are presented mostly by their analytic approach, in “steps”, and no specific hypotheses are presented. A “theory” is broader than a hypothesis.

Methods

I am hesitant to call these data “nationally representative” since they were drawn from an industry-sponsored web panel, and no description of the sampling frame or sampling methods are provided. “Quota sampling” is not representative if every potential participant does not have a known probability of participation. Further, no response rates were provided that would support calling the samples representative. I suggest providing a more clear description of how the samples were collected, and urge caution in calling these samples nationally representative.

Relatedly, I am skeptical that one could exclude 60% of a sample and it would be MCAR. Also, no description of what variables had the missing data was provided, and this information would be helpful for understanding the dataset better. Similarly, it would be helpful to know how the two datasets compared. There is a description in the supplementary materials, but it would be easier for the reader to understand this if it was in the text. This is especially important since there does seem to be some differences by age between the two samples.

The PCA analysis should be described in the measures section. It seemed to come out of nowhere in the results section.

It would be helpful if the precise model specifications used for each dataset are described. While there was a lot of detail provided about the statistical approach, the precise variables used in the model seemed to get lost in all the detail.

I think the measure for “mental health” needs to be better described, and described consistently throughout the manuscript. As written, particularly in the introduction, as though mental health as a psychological construct was measured, and this is not the case. Rather, a subjective measure of how much someone perceived the pandemic to impact their mental health and well-being was measured, and this is very different. A phrase like “perceived mental health impact” or something to that effect would be more accurate.

Also, I encourage the authors to consider the utility of this measure at all, given its limitations. I wonder if an analysis that simply looked at differences in the relationships between drinking motives and alcohol use between the two datasets might be more informative than one that includes a limited measure of mental health that could well muddle the utility of the model.

Results

I found the results as structured difficult to follow. I suggest structuring them by aim rather than by statistical approach.

Discussion

I think more discussion is warranted on the validity of using SEM when such important assumptions are not met, that is, that there were no omitted confounders, which seems very unlikely.

6. PLOS authors have the option to publish the peer review history of their article (what does this mean?). If published, this will include your full peer review and any attached files.

Reviewer #1: No

Reviewer #2: No

---

## [Author Response · Author response to Decision Letter 0]

20 Oct 2022

PONE-D-22-17576

UK alcohol consumption during the COVID-19 pandemic: The role of drinking motives, employment and mental health concerns

Dear editor and reviewers

Thank you very much for the constructive feedback, which have benefited the paper. Below we outline how we have addressed each of your comments and hope that, having done so, you will now find the paper suitable for publication in PLoS One.

Yours sincerely

Rebecca

Editor 

The data come from an industry sponsored source and although the authors suggest that they are 'nationally representative', no supporting data are provided. 

**We have changed our phrasing in response to R2’s comment on this.

Comparisons are made between cross-sectional data collected pre-pandemic (2018) and then during the pandemic in 2020, to assess 'changes'. This is a major weakness of the study, as longitudinal follow up would have been the appropriate method.

**To address this concern we have added the following to the discussion: :

“In contrast, the current work assessed differences in population-level drinking patterns between time points. Of course, future research will need to take a longitudinal approach to better assess whether drinking motives, for instance, caused within-persons, between-persons, and ultimately population-level changes in alcohol consumption drinking patterns between time points.”

We have also retained a second paragraph which outlines the limitations of our current model assumptions and the need for longitudinal research in the future.

 It is also noted that 5,248 (58%) participants were excluded from the 2020 dataset and 1,004 (11%) from the 2018 dataset due to missingness. The implications of this loss of information which disproportionately affects persons evaluated during the pandemic has not been addressed. 

** We have added an acknowledgement of this pertinent point to the discussion, where we also tackle R2’s point about the framing of this sampling as “nationally representative”.

The paper is verbose and meanders, and as such, its style can be significantly improved. The Introduction can be markedly reduced from the current 6 pages. 

**In response to this and the comments from R2, the introduction has been edited and reduced in length.

There are multiple statistical analyses (model fit statistics; indirect effect values; invariance tests) which do not contribute to the overall hypothesis and would be more appropriately included with the Supplementary materials. Removing these tables would improve the focus of the paper. 

*We have removed a number of tables and placed them into the supplementary materials document. These have been relabelled/numbered accordingly and are referred to in the main text of the document (for easy indexing by the reader).

Similarly the Discussion is too long and also lacks clarity and focus. 

** We have edited and reduced the length of the discussion accordingly, while also making additions to respond to reviewer comments.

**We have uploaded our data with this resubmission 

Reviewer #1: 

The article is well written and clear. It would be good to discuss other factors that might be related to increased consumption, such as increased alcohol availability at home, which in turn are known factors to lead to more consumption. Time spent online during the pandemic could be another factor related to increased consumption, but it was never asked. 

** Thanks for this suggestion, we have now added this as an avenue for future direction in the discussion. 

In addition, the authors can better explain how an online survey becomes representative of the general population, when there are differences in access to internet, mobile phones, etc. **In response to this and comments from R2, we have emphasised the quota sampling approach that was used to obtain this sample. We have also a line in the discussion about how the survey may have been limited in terms of its representation. We also highlight access issues for those who do not have access to technology.

Finally, the data from 2020 reflect the beginning of the pandemic and not its peak, so this limitation can also be discussed, as we need to have more information afterwards and after the pandemic, to better contextualize the findings and their robustness.

**We have added this to the study future directions, thanks for the suggestion.

Reviewer #2: 

Overall

Throughout the manuscript the authors state that they are looking at “change” in alcohol consumption as a result of the pandemic. This not possible given that the data they used is not cohort data. The data allows them to examine *differences* between pre and early pandemic periods, but they cannot assess changes in either alcohol use or in drinking motives without following the same people over time. I suggest reframing the manuscript to reflect this.

** We have reframed as you suggest.

Introduction

The introduction does well to provide background on the drinking motives and alcohol use during the pandemic, but it was difficult to follow because it seemed wander between a global perspective and then a UK one. It was hard to tell when the intro was discussing findings that were specific to the UK and when they were from other countries. If putting the UK in a global framework is important to the authors, then I suggest starting with a global perspective then honing in on the specific UK situation.

**Thanks for pointing this out, we have framed the intro with a closer focus on the UK, with clearer emphasis when we are drawing on studies from outside the UK to reflect on how patterns of behaviour are the same/different to those observed in the UK. 

Also, when the authors note different findings across different countries and the need for additional research to address these differences, the implication is that findings should be consistent across countries. This seems unnecessary since we would expect different reasons for drinking across different cultural contexts. Rather, it would seem that to gain a full picture of the impact of the pandemic on drinking globally, we want rigorous research from all these areas, and not that the findings all have to agree with each other.

**We agree and have altered our framing in the intro accordingly, in response to this and other reviewer’s comments.

No clear reason for why we need this research is presented, that is, how the findings will be used or what they will inform. This is admittedly a pet peeve of mine, but the argument that this work is necessary to “fill the gaps” without explaining why we need to fill these gaps in the first place is a less-than-compelling argument for undertaking this work.

**We have emphasized the intervention-based importance of our findings, while remaining mindful of the editorial instruction to shorten the length of the introduction. 

No theoretical framework for examining drinking motives is presented, such as the self-medication hypothesis. Also, the frequent comorbidity between alcohol problems and mental health is not well described in the introduction.

**We take your point here. Yet, we were under instruction from the editor to shorten the length of the introduction, which we have now done. Adding substantial detail would lengthen this again. As such, we have compromised with a brief note on the self-medication hypothesis and hope you will find this sufficient. We would also take editorial instruction if more info is felt necessary.

Finally, the aims are unclear and confusing as written. They are presented mostly by their analytic approach, in “steps”, and no specific hypotheses are presented. A “theory” is broader than a hypothesis.

** This research was exploratory and thus we did not specify any hypotheses. We feel that it would be disingenuous to now add these in, knowing the results as we do. We have however, more clearly linked our aims with the statistical steps we outline. As follows:

“Our approach consists of two steps with associated aims. Step 1 aimed to examined between-persons differences in alcohol consumption to explore factors that may explain differences in self-reported alcohol consumption. As such, here we examine correlates of pandemic drinking during a period of COVID-19 related social restrictions (between 27 August and 15 September 2020; final n = 3,798) and, using structural equation modelling (SEM), test the theory that COVID-19-related employment and the perceived impact of COVID-19 on one’s mental health (subjective mental health) impact self-reported alcohol consumption (AUDIT-C) via drinking motives. Step 2 aimed to explore the mean differences in consumption at the population-level (between two timepoints) and assess whether these may be driven by specific factors. Accordingly, here, we move beyond previous research largely based on single-study cross-sectional designs by leveraging UK nationally representative data gathered before and during the pandemic to (a) test for heterogeneity in pandemic-occasioned change in alcohol consumption, as well as (b) test the theory that change in drinking motives may explains pandemic-occasioned differences in consumption.”

Methods

I am hesitant to call these data “nationally representative” since they were drawn from an industry-sponsored web panel, and no description of the sampling frame or sampling methods are provided. “Quota sampling” is not representative if every potential participant does not have a known probability of participation. Further, no response rates were provided that would support calling the samples representative. I suggest providing a more clear description of how the samples were collected, and urge caution in calling these samples nationally representative.

**~Thanks for this point. We do not have data on response rates from the provider but in response to your comment, we have changed our phrasing throughout to emphasise that this was quota sampled data. We have also added a line to the limitations section, noting the issues of this approach, as you note here. The following information has also been added to the “study population” subheading (along with a reference to the technical report of the most recent survey): “Data were obtained using a quota sampling approach designed to collate representative samples. YouGov’s research panel consists of over 1,000,000 people in the UK. For this research, members were selected based on known demographic characteristics (specifically age, gender, social grade and region), and the sample surveyed was representative of the four countries of the UK. More specifically, predefined quotas were obtained based on the known population profile of adults aged 18-85 years according to gender, age, social grade, and region. The final data were weighted to reflect this profile (Newbold, 2021).”

Relatedly, I am skeptical that one could exclude 60% of a sample and it would be MCAR. Also, no description of what variables had the missing data was provided, and this information would be helpful for understanding the dataset better. 

***We appreciate your concern and have doublechecked whether the data was missing at random using Little’s MCAR test. We did this for the 2018 data and 2020 data separately and found that both were missing at random. We have also added text to specify which variables were missing data to show how the final samples were arrived at.

Similarly, it would be helpful to know how the two datasets compared. There is a description in the supplementary materials, but it would be easier for the reader to understand this if it was in the text. This is especially important since there does seem to be some differences by age between the two samples.

** This information is in now in the text at the bottom of page 17 (under title Step 2: 2020 vs 2018 data) and in Table 4.

The PCA analysis should be described in the measures section. It seemed to come out of nowhere in the results section.

** So moved 

It would be helpful if the precise model specifications used for each dataset are described. While there was a lot of detail provided about the statistical approach, the precise variables used in the model seemed to get lost in all the detail.

** Having moved the PCA analyses to the measures section, as suggested, we believe that our model specifications are now clearer and these are listed under the following sub headings (Step 1: 2020 data; Step 2: 2020 vs 2018 data).

I think the measure for “mental health” needs to be better described, and described consistently throughout the manuscript. As written, particularly in the introduction, as though mental health as a psychological construct was measured, and this is not the case. Rather, a subjective measure of how much someone perceived the pandemic to impact their mental health and well-being was measured, and this is very different. A phrase like “perceived mental health impact” or something to that effect would be more accurate.

** We thank you for this observation. We have changed the phrasing to emphasise that we mean the perceived impact of COVID-19 on one’s mental health (labelling the variable as subjective mental health). We have also changed the title to “subjective mental health” so as not to mislead.

Also, I encourage the authors to consider the utility of this measure at all, given its limitations. I wonder if an analysis that simply looked at differences in the relationships between drinking motives and alcohol use between the two datasets might be more informative than one that includes a limited measure of mental health that could well muddle the utility of the model.

**We have added a note on the potential limitation of this variable and suggest future research deploy a standardised MH assessment. We nevertheless think the results with this measure are interesting and respectfully argue for its retainment.

Results

I found the results as structured difficult to follow. I suggest structuring them by aim rather than by statistical approach.

**We have now linked our aims with our statistical approach, for clarity.

Discussion

I think more discussion is warranted on the validity of using SEM when such important assumptions are not met, that is, that there were no omitted confounders, which seems very unlikely.

**We had already acknowledged that our models assumed that there were no omitted confounders, yet we did not have design features like experimental control that could provide strong assurance this assumption held. As such we have no evidence that our assumptions were either met or not met and all we can do is acknowledge that fact and say, as we have, that longitudinal designs would be an improvement (though they also will have untested assumptions like no confounders at the within-persons level). However, to better articulate our point, in line with your comment, we have now revised the section of the discussion on our model assumptions. It reads as follows:” Our models assumed, for instance, that there were no omitted confounders, yet we did not have design features like experimental control that could provide strong assurance this assumption held. Although the lack of differences in the effects of drinking motives between the levels of inter-individual and time point differences is consistent with causation, it remains possible that our estimates are biased by omitted variables that also have effects similar in magnitude at both these levels. Thus, causal inference from our results must remain relatively tentative. Because experimental tests of the theories evaluated here may be infeasible, future research adding a longitudinal perspective seems to provide a logical next step in terms of conducting a more rigorous test of our assumptions. Research tracking longitudinal variability in individual drinking practices would add further insights into how the changing nature of the pandemic over time (i.e. in terms of divergences in the degree/nature of social restrictions etc over time) may also have impacted drinking practices and thus provide insights into within-person changes in drinking. Again, we do see this study as a significant contribution to the current literature, which is largely based on smaller studies of data collected at only one time points; however, we also recognize that longitudinal designs can better address confounding in future research.”

---

## [Editor Report · Decision Letter 1]

2 Jan 2023

PONE-D-22-17576R1UK alcohol consumption during the COVID-19 pandemic:  The role of drinking motives, employment and subjective mental healthPLOS ONE

Dear Dr. Monk,

Thank you for submitting your manuscript to PLOS ONE. After careful consideration, we feel that it has merit but does not fully meet PLOS ONE’s publication criteria as it currently stands. Therefore, we invite you to submit a revised version of the manuscript that addresses the points raised during the review process.

There are edits that are still required. Please do a detailed grammatical review of the entire article making appropriate amendments, some of the areas of concern noted have been highlighted below:

P3: Introduction: ... alcohol-related behaviours (rather than alcohol behaviours);

P5: clarify the following which is not clear (although see Tovmasyan et al. 2022);

P5. Please Correct 'çomparatively less advantaged...' to ... çomparatively disadvantaged.....'

P5. Please Correct: ..Çovid-19 also lead ... ' to '... Covid-19 also led..'.

P6: Please Correct: ..'..are also variables factors...'

P6: Please Correct: ..'towards filling these gaps and providing EVIDENCE upon which to develop ...'

P8: Please Correct: ..'liker scale'.... to 'Likert scale'...

P8: Please Correct punctuation at end of this specific paragraph

P21: Please Correct: the following phrase - '.. coping and social motives, which were in turn, were associated with elevated alcohol use....' -

The term 'dataset' has not been spelt the same way throughout the text.

We look forward to receiving your revised manuscript.

Kind regards,

Anselm J. M. Hennis

Academic Editor

PLOS ONE
---

## [Author Response · Author response to Decision Letter 1]

4 Jan 2023

PONE-D-22-17576

UK alcohol consumption during the COVID-19 pandemic: The role of drinking motives, employment and mental health concerns

Dear editor and reviewers

Thank you very final round of comments on the paper. Below we outline how we have addressed each of your comments and hope that, having done so, you will now find the paper suitable for publication in PLoS One.

Yours sincerely

Rebecca

There are edits that are still required. Please do a detailed grammatical review of the entire article making appropriate amendments.

** We have now completed a through proofread, and have addressed the below comments also.

some of the areas of concern noted have been highlighted below:

P3: Introduction: ... alcohol-related behaviours (rather than alcohol behaviours)

**So changed 

P5: clarify the following which is not clear (although see Tovmasyan et al. 2022);

**The following has been added “which suggests that a model of affect intensity regulation may be a more advantageous way of understanding the alcohol-mood nexus, rather than theorising about mood valence”

P5. Please Correct 'çomparatively less advantaged...' to ... çomparatively disadvantaged.....'

** Changed

P5. Please Correct: ..Çovid-19 also lead ... ' to '... Covid-19 also led..'.

** Changed

P6: Please Correct: ..'..are also variables factors...'

** Corrected

P6: Please Correct: ..'towards filling these gaps and providing EVIDENCE upon which to develop ...'

** Added

P8: Please Correct: ..'liker scale'.... to 'Likert scale'...

**So changed

P8: Please Correct punctuation at end of this specific paragraph

** Addressed 

P21: Please Correct: the following phrase - '.. coping and social motives, which were in turn, were associated with elevated alcohol use....' 

** Corrected

The term 'dataset' has not been spelt the same way throughout the text.

**Changed to dataset throughout

---

## [Editor Report · Decision Letter 2]

6 Mar 2023

UK alcohol consumption during the COVID-19 pandemic:  The role of drinking motives, employment and subjective mental health

PONE-D-22-17576R2

Dear Dr. Monk,

We’re pleased to inform you that your manuscript has been judged scientifically suitable for publication and will be formally accepted for publication once it meets all outstanding technical requirements.

Kind regards,

Anselm J. M. Hennis

Academic Editor

PLOS ONE
---

## [Editor Report · Acceptance letter]

17 Mar 2023

PONE-D-22-17576R2 

UK alcohol consumption during the COVID-19 pandemic:  The role of drinking motives, employment and subjective mental health 

Dear Dr. Monk:

I'm pleased to inform you that your manuscript has been deemed suitable for publication in PLOS ONE. Congratulations! Your manuscript is now with our production department. 

Kind regards, 

on behalf of

Dr. Anselm J. M. Hennis 

Academic Editor

PLOS ONE